# The Anti-Cholinesterase Potential of Fifteen Different Species of *Narcissus* L. (Amaryllidaceae) Collected in Spain

**DOI:** 10.3390/life14040536

**Published:** 2024-04-22

**Authors:** Luciana R. Tallini, Giulia Manfredini, María Lenny Rodríguez-Escobar, Segundo Ríos, Vanessa Martínez-Francés, Gabriela E. Feresin, Warley de Souza Borges, Jaume Bastida, Francesc Viladomat, Laura Torras-Claveria

**Affiliations:** 1Departament de Biologia, Sanitat i Medi Ambient, Facultat de Farmàcia i Ciències de l’Alimentació, Universitat de Barcelona, Av. Joan XXIII 27-31, 08028 Barcelona, Spain; 2Faculdade de Farmácia, Universidade Federal do Rio Grande do Sul, Av. Ipiranga 2752, Porto Alegre 90610-000, RS, Brazil; 3Dipartimento di Scienze della Vita, Universita’ Degli Studi di Modena e Reggio Emilia, 41121 Modena, Italy; 4Estación Biológica Torretes-Jardín Botánico de la UA, Universidad de Alicante, Crtra. Sant Vicent del Raspeig, s/n, 03690 Alicante, Spain; 5Department Biología Aplicada, Area de Botánica, Universidad Miguel Hernández, Av. Universidad, s/n, 03202 Elche, Spain; 6Instituto de Biotecnología, Facultad de Ingeniería, Universidad Nacional de San Juan, Av. Libertador General San Martin 1109 Oeste, San Juan 5400, Argentina; 7Laboratory of Natural Products, Department of Chemistry, Federal University of Espirito Santo, Vitória 29075-910, ES, Brazil

**Keywords:** acetylcholinesterase, Amaryllidaceae, alkaloids, Alzheimer’s disease, butyrylcholinesterase, *Narcissus*

## Abstract

*Narcissus* L. is a renowned plant genus with a notable center of diversity and is primarily located in the Mediterranean region. These plants are widely recognized for their ornamental value, owing to the beauty of their flowers; nonetheless, they also hold pharmacological importance. In Europe, pharmaceutical companies usually use the bulbs of *Narcissus pseudonarcissus* cv. Carlton to extract galanthamine, which is one of the few medications approved by the FDA for the palliative treatment of mild-to-moderate symptoms of Alzheimer’s disease. The purpose of this study was to evaluate the potential of these plants in Alzheimer’s disease. The alkaloid extract from the leaves of different species of *Narcissus* was obtained by an acid-base extraction work-up -procedure. The biological potential of the samples was carried out by evaluating their ability to inhibit the enzymes acetyl- and butyrylcholinesterase (AChE and BuChE, respectively). The species *N. jacetanus* exhibited the best inhibition values against AChE, with IC_50_ values of 0.75 ± 0.03 µg·mL^−1^, while *N. jonquilla* was the most active against BuChE, with IC_50_ values of 11.72 ± 1.15 µg·mL^−1^.

## 1. Introduction

*Narcissus* L. is a well-known plant genus that belongs to the Amaryllidaceae family, specifically within the Amaryllidoideae subfamily [1]. These plants, commonly referred to as daffodils, are highly favored in gardens and serve as a significant commercial crop [2]. This genus encompasses around 100 wild species, primarily concentrated in southwestern Europe, with a significant center of diversity in the Iberian Peninsula—where 90% of all species are present, especially in Spain, and extend throughout North Africa [2,3,4].

The native habitats of the *Narcissus* species exhibit remarkable diversity, encompassing a broad spectrum of landscapes that span from lowland to mountainous regions. This genus includes a rich tapestry of environments, including verdant grasslands, rugged scrublands, serene woodlands, meandering riverbanks, and secluded rocky crevices [2]. The presentation of blooms exhibited by the majority of these species is a characteristic feature observed during late winter and spring. However, there are a limited number of species that deviate from this pattern and blossom during the autumnal season [5,6].

The hybridization of some of these species has led to the development of commercial *Narcissus* cultivars, which, in most instances, are larger and more robust than their wild ancestors [2,7]. This genus stands out as the main choice among commercial bulb planters, showcasing remarkable importance in various horticultural and commercial contexts [8] in Western Europe, being cultivated since the 16th century in the Netherlands. Currently, the United Kingdom, the Netherlands, and the United States are the main producing countries of *Narcissus* bulbs. Until the late nineties, *Narcissus* bulb production was oriented only towards ornamental use, but since 1999, the purpose of extracting galanthamine from these bulbs started to play an important and growing role [9]. Therefore, although the cultivation of *Narcissus* for ornamental purposes has had a long tradition, leading to extensive knowledge of its growing and breeding, as well as the production of large stocks of raw materials, only a few of the numerous cultivars developed could be cultivated in significant amounts and quality for chemical extraction [2]. This question has been successfully solved by in vitro cultivation [10].

The medicinal properties of the *Narcissus* species have been documented in the literature for many years. Hippocrates of Kos (460-370 BCE), the father of modern medicine, advocated the use of oil from the *Narcissus* species to alleviate symptoms that today would be recognized as cancer [11]. Dioscorides, in the 1st century, mentions some of their medicinal properties due to the presence of alkaloids, as well as the sedative and narcotic effects of their aroma [12]. In recent decades, the genus *Narcissus* has provided a range of useful or potentially valuable compounds, of which galanthamine has been extensively studied. Galanthamine hydrobromide is one of the few medicines available used for the palliative treatment of symptoms derived from Alzheimer’s disease, owing to its ability to inhibit the enzyme acetylcholinesterase. This alkaloid—specifically Amaryllidaceae alkaloid—was first isolated in 1952 from the species *Galanthus woronowii* and received the FDA’s approval in 2001 [13,14]. Pharmaceutical companies concentrated their search for galanthamine on the Amaryllidoideae subfamily, as it is the exclusive natural source of this metabolite [2,15]. For *Narcissus* cultivars, different studies have been carried out on variations in galanthamine content in bulbs, depending on their geographical origin or changes due to the addition of fungicides [16,17]. Torras-Claveria and co-workers [18] investigated the galanthamine content and bioactivity of more than one hundred *Narcissus* cultivars, highlighting those with higher galanthamine content and higher acetylcholinesterase inhibitory activity. In Central and Western Europe, the extraction of galanthamine is mainly derived from *Narcissus pseudonarcissus* cv. Carlton. Meanwhile, in Eastern Europe, *Leucojum aestivum*, known as snowflakes, serves as a key source, and it has been used in the past, although its collection for this purpose is currently not possible, as this species is in decline. In China, the red-tubed lily, *Lycoris radiata*, contributes to galanthamine production as well [9].

Amaryllidaceae alkaloids are unique structures originating from the Amaryllidoideae subfamily, covering a diverse group of compounds that exhibit interesting biological properties. Currently, there are more than 650 structures of Amaryllidaceae alkaloids reported in the literature [19]. While significant progress has been made in recent years, the comprehensive exploration of the diversity of structures, reactions, and genes within the Amaryllidaceae family remains ongoing. All of these metabolites originate from the aromatic amino acids L-phenylalanine (L-Phe) and L-tyrosine (L-Tyr), which, through distinct enzymatic reactions, ultimately lead to the formation of norbelladine [20,21]. The enzyme norbelladine 4′-*O*-methyltransferase catalyzes the production of 4′-*O*-methylnorbelladine, which is considered the main common precursor to Amaryllidaceae alkaloids [21]. The oxidative phenolic coupling of 4′-*O*-methylnorbelladine leads to the three main skeleton types that form the bases for the extensive structural diversity of Amaryllidaceae alkaloids. The *ortho-para’* coupling of 4′-*O*-methylnorbelladine results in the formation of the lycorine- and homolycorine-type skeletons, *para-para’* originates the crinine-, haemanthamine-, tazettine-, narciclasine- and montanine-type structures, and *para-ortho’* gives the galanthamine-type skeleton [22]. Since the initial isolation of the alkaloid lycorine from *Narcissus pseudonarcissus* in 1877, significant advancements have been achieved in the study of Amaryllidaceae plants. Nevertheless, they continue to represent a relatively underexplored source of phytochemicals [3]. Generally, within each plant, a variety of related alkaloids is present, consisting of a few dominant metabolites and multiple minor compounds with varying substituent positions [3].

The historical bond between Amaryllidaceae and medicine stands as a testament to the enduring allure and profound significance of these remarkable compounds within the realm of human health and well-being. In the present day, continued research into Amaryllidaceae alkaloids promises to unveil further insights into their potential applications and therapeutic benefits. In this way, the aim of this study was to describe the potential different wild species of *Narcissus* L. collected in Spain against Alzheimer’s disease through cholinesterase inhibition assays.

## 2. Materials and Methods

### 2.1. Plant Material

Fifteen different species of *Narcissus* L. were provided from the Iberian *Narcissus* Collection of the Torretes Biological Research Station—Botanical Garden of the UA, in Alicante, Spain. All the samples were collected during the flowering season, April 2023. The origins of the species are different localities of the Iberian Peninsula, as described in Table 1 and Figure 1. All the species were authenticated by botanists Dr. Segundo Ríos and Dr. Vanessa Martínez-Francés. Most wild daffodils are distributed in small, scattered populations, in very specific and fragile microhabitats such as wetlands, rocky areas, and deciduous forests [2,23,24,25,26]. These populations with very small numbers (less than 1000 individuals) have a small surface area (often less than 1 ha), making them extremely vulnerable to changes due to natural (climate change, predation, etc.) or anthropogenic causes (land use, grazing, collecting, etc.), which endanger their survival [27,28].

The ecology of many species from the Pseudonarcissus DC. section, as *N. asturiensis*, *N, jacetanus*, *N. minor*, and *N. vasconicus* are perennial grasslands of external fringe woodlands. Other species, such as *N. bujei* and *N. genesii-lopezii*, are present in perennial mesophytic grasslands, and *N. confusus is* present in the secondary mesoforests of deciduous oaks (Table 1).

More ecological diversity is observed in the Nevadensis Zonn. section. *N. nevadensis* grows in the secondary mesoforests of deciduous oaks and the perennial grasslands of external fringe woodlands. The existing populations of *N. alcaracensis* develop on lacustrine and helophyte communities dominated by *Carex hispida*, and *N. yepesii* is present in meadows and chionophilous grassland vegetation (Table 1).

*N. jonquilla*, from section Jonquillae DC., grows in lacustrine and riverine helophyte graminoid communities, while *N. assoanus* grows in perennial succulent grassland in rocky, open communities. The representative of the Bulbocodium section, *N. hedraeanthus*, is present in meadows and chionophilous grassland vegetation. From the Ganymedes section, *N. pallidulus* grows in perennial mesophytic grasslands and bare sandy soils, and *N. tazetta* of section Tazettae DC. has been collected from gardens where it was cultivated (Table 1). Although ornamental domestication of the genus *Narcissus* by the United Kingdom and Holland began in the 16th century [7,24], today there is still continuous extraction of wild bulbs in Spain, Portugal, and North Africa to satisfy the global private collecting market, to obtain new disease-resistant genes in commercial bulbs, and to satisfy the demand for galanthamine by pharmaceutical companies. Despite the scarce data on the populations and conservation status of most of the species, the information collected for some of them has allowed for their inclusion in the IUCN Red List, considered a critical indicator of the health of biodiversity. Two species of section Nevadensis Zonn. Analyzed in this work, *N. alcaracensis* and *N. nevadensis*, and another of section Pseudonarcissus DC., *N. bujei,* have been assessed as Endangered (EN) in the IUCN list [30,31,32]. Another species of this first mentioned section, *N. yepesii*, has been evaluated and classified as Vulnerable (VU) [33].

*N. asturiensis* and *N. hedraeanthus* are listed under Least Concern (LC), with no other major conservation measures recommended [34,35]. *N. pallidulus,* although not mentioned, is included in *N. triandrus*, which is assessed as Least Concern (LC), considering its wide distribution, with large and stable populations; it is unlikely that existing threats will cause them to seriously decrease in the near future [36]. However, it should be added that the different species of Section Ganymedes (Salisb.) Schultes f. studied in [37] have not been taken into consideration in the preparation of this manuscript, and different problems are present regarding each of them, both with respect to population size and natural and anthropogenic issues, requiring, therefore, a new review.

Only the leaves were collected for biological assays. The use of leaves instead of bulbs for biological activity studies ensures the maintenance of the Iberian Daffodil Collection, allowing for its reproduction. Moreover, depending on the phenological stage of the plant, the alkaloid content and, therefore, the biological activity, may vary from bulb tissue to leaf tissue, and sometimes leaves may contain more alkaloids and be more active than bulbs [18,38].

### 2.2. Alkaloid Extracts Preparation

For the evaluation of the bioactivity, purified alkaloid extracts were performed. The species were meticulously processed as follows: First, they were cut into pieces and dried at a controlled temperature of 40 °C. Afterward, the dried plant material was finely powdered using a rotary blade mill, specifically a stainless-steel grinder (Taurus, Oliana, Spain). To extract the desired compounds, 1 g of the resulting powder was subjected to a maceration process with methanol at 25° for three days. During this period, the solvent was replaced each day using 3 × 50 mL aliquots and submitted to an ultrasonic bath (20 min, 4 daily intervals). Following methanolic extraction, the mixture was meticulously strained, and the solvent was carefully evaporated under reduced pressure, leaving behind crude extracts. These crude extracts were subsequently acidified using 30 mL of a 2% (*v*/*v*) sulfuric acid solution, lowering the pH to 2. After acidification, an ethyl acetate treatment (using 3 × 50 mL) was employed to eliminate neutral materials. The next step involved adjusting the pH of the remaining aqueous solution to a range of 9–10, achieved by the addition of a 25% (*v*/*v*) ammonium hydroxide solution. The volume of the ammonium hydroxide solution employed was the quantity required to achieve the aforementioned pH. Finally, the alkaloids were extracted with ethyl acetate (using 3 × 50 mL). After evaporation of the solvent, the dried alkaloid extract (AE) was obtained.

### 2.3. Cholinesterase Inhibitory Activity

Levels of ACh and BuCh decrease in patients with Alzheimer’s disease. Inhibition of AChE and BuChE has been shown to maintain ACh and BuCh levels in the brain, reducing disease progression [13,39]. The activity of *Narcissus* extracts inhibiting these enzymes can be assessed with colorimetric AChE and BuCHE inhibition assays, based on the formation of thiobenzoate anion (yellow) following the reaction of thiocholine and 2,2′-dinitro-5′-dinitro-5′-dithiobenzoic acid when the enzymes are active.

The inhibition activity of AChE and BuChE was assessed using the method described by Ellman and co-workers [40], with certain adaptations, as outlined by López and co-workers [41]. Enzyme stock solutions containing 518U of AChE from *Electrophorus electricus* (Merck, Darmstadt, Germany) and BuChE from equine serum (Merck, Darmstadt, Germany) were prepared and subsequently stored at −20 °C. 5,5-Dithiobis (2-nitrobenzoic acid) (DTNB), *S*-butyrylthiocholine iodide (BTCI), and acetylthiocholine iodide (ATCI) were supplied by Merck (Darmstadt, Germany). The reaction was initiated by mixing 50 µL of AChE or BuChE (both enzymes were employed at a concentration of 6.24 U in phosphate buffer (8 mM K_2_HPO_4_, 2.3 mM NaH_2_PO_4_, 0.15 NaCl, pH 7.5)) and 50 µL of the alkaloid extract dissolved in the same buffer solution. Immediately, the plates were incubated for 30 min at 25°. Finally, 100 µL of the substrate solution (comprising 0.1 M Na_2_HPO_4_, 0.5 M DTNB, and 0.6 mM ATCI or 0.24 mM BTCI in Millipore water, adjusted to pH 7.5) were introduced. Ten minutes later, the absorbance was measured at 405 nm using a Labsystem microplate reader (Helsinki, Finland). The activity of the enzymes was estimated as percentages in relation to a control (which consisted of a buffer without any inhibitor). Galanthamine served as a positive control. The galanthamine concentrations used for this positive control were the following: 0.1, 0.2, 0.3, 0.4, 0.5, 1.0, and 2.0 µg·mL^−1^ for AChE; and 1, 4, 6, 8, 10, 12, and 15 µg·mL^−1^ for BuChE. The calibration curves of samples A (0.1, 1.0, 3.0, 5.0, 7.0, 10, and 15 µg·mL^−1^), B, E and L (0.05, 0.1, 0.5, 1.0, 3.0, 5.0, and 10 µg·mL^−1^), C (0.1, 0.5, 1.0, 2.0, 3.0, 4.0, and 5.0 µg·mL^−1^), D and G (1.0, 2.5, 5.0, 7.5, 10, 15, and 25 µg·mL^−1^), F (5.0, 10, 15, 25, 50, 75, and 100 µg·mL^−1^), H (5.0, 10, 15, 20, 25, 50, and 75 µg·mL^−1^), I (1.0, 2.0, 3.0, 4.0, 5.0, 7.0, and 10.0 µg·mL^−1^), K (1.0, 10, 25, 50, 75, 100, and 125 µg·mL^−1^), M and O (0.1, 0.5, 1.0, 3.0, 5.0, 7.0, and 10.0 µg·mL^−1^), and N (5.0, 7.5, 10, 15, 25, 50, and 75 µg·mL^−1^) were applied to obtain the IC_50_ values against the AChE enzyme. To obtain the IC_50_ values against BuChE, the following curves were used: C and K (10, 25, 50, 75, 100, 125, and 150 µg·mL^−1^), E (1, 5, 10, 15, 25, 50, and 75 µg·mL^−1^), G, H, I, L, M and O (1, 10, 35, 50, 75, 100, and 125 µg·mL^−1^), and N (10, 25, 50, 100, 125, and 150 µg·mL^−1^). Analysis of the cholinesterase’s inhibitory data was conducted using Prism 10 software.

### 2.4. Statistical Evaluation

The inhibition of the cholinesterase activity of the *Narcissus* species was assessed using three separate assays. The PRISM software was used to analyze the results. The data are presented as the average ± standard deviation (SD). The significance of the results is represented versus the control (Gal), and it is indicated as follows: **** *p* < 0.0001, *** *p* < 0.001, ** *p* < 0.01, and ns (not significant). A one-way ANOVA test was performed following Dunnet’s multiple comparison test, comparing the differences with respect to the outcome of galanthamine with both AChE and BuChE.

## 3. Results and Discussion

The alkaloid extracts from all the *Narcissus* species collected in Spain were obtained through acid-base extraction, as described in Section 2.2. The yield of each species is available in Table 2. The average income value was 1.32%, with the highest value at 6.14% (sample N) and the lowest at 0.47% (sample B), which represent the species *N. yepesii* and *N. jacetanus*, respectively.

In vitro assessments were carried out to examine the inhibitory potential of the fifteen alkaloid extracts from *Narcissus* leaves against the enzymes AChE and BuChE. Among these plant samples, fourteen demonstrated activity against AChE, while only nine exhibited activity against BuChE (see Figure 2 and Figure 3). As was expected, all the species evaluated herein presented better results against AChE than BuChE. The species *N. jacetanus* (sample B) showed the best inhibition values against AchE, with IC_50_ values of 0.75 ± 0.03 µg·mL^−1^, while *N. jonquilla* (sample L) was the most active against BuChE, with IC_50_ values of 11.72 ± 1.15 µg·mL^−1^. The species *N. jacetanus* and *N. jonquilla* are illustrated in Figure 4 and Figure 5, respectively. The samples *N. assoanus*, *N. minor*, *N. confuses*, and *N. jonquilla* also presented noteworthy activity against AChE, with values of IC_50_ of 0.99 ± 0.06, 0.81 ± 0.10, 1.04 ± 0.07, and 1.88 ± 0.05 µg·mL^−1^, respectively. Regarding BuChE inhibition activity, *N. confusus* and *N. genesii-lopezii* also presented high and relevant activity, with IC_50_ of 12.83 ± 0.87 and 11.98 ± 0.93 µg·mL^−1^, respectively. Thus, *N. jonquilla* stands out for being the species of *Narcissus* with the highest global cholinesterase activity inhibition, followed by *N. confusus* and *N. jonquilla*.

Considering their biological potential, it is important to account for the potential synergistic interactions among Amaryllidaceae alkaloids in plant extracts [42]. These interactions, previously reported to contribute to acetylcholinesterase (AChE) inhibition, should be taken into consideration when explaining the anticholinesterase potential of certain Amaryllidaceae species [42].

Among other samples, Havlasová and co-authors [43] evaluated the inhibitory potential of *N. jonquilla* var. *henriquesii* against AChE and obtained IC_50_ values of 32.6 ± 4.3 µg·mL^−1^, while for galanthamine, it was 1.7 ± 0.06 µg·mL^−1^ [43]. As documented in the literature, the galanthamine-type skeleton, specifically the alkaloids galanthamine and sanguinine, are commonly active against AChE and BuChE [41]. In the literature, a great amount of galanthamine-type structures is reported, such as galanthamine, lycoramine, and narwedine in the species *N. jonquilla*, representing about 65% of its alkaloid profiling [4,44]. Furthermore, haemanthamine, tazettine, jonquailine, and narciclasine- and lycorine-type structures were also described in this plant species [4,44,45]. Furthermore, the majority of the reported alkaloids of *N. jonquilla* have been studied in terms of molecular modelling in front of AChE and BuChE, and data reported in the literature support the fact that they could be responsible for the especially high activity of this species inhibiting BuChE and AChE. Galanthamine has been reported to have good docking scores for BuCHE and ACHE [46,47,48,49,50]. Lycoramine has been reported to have molecular docking values of −8.84, −9.08, −8.87, −8.64, and −8.41 kcal·mol^−1^ for the human acetylcholinesterase X-ray crystals 4EY5, 4EY6, 4Ey7, 4M0E, and 4M0F, respectively, while values for galanthamine were −8.59, −8.75, −9.83, −7.90, and −8.74 kcal·mol^−1^, according to Tallini et al., 2022 [50]. Narwedine has been reported to have better scores (−9.15, −9.70, −10.41, −8.69, and −9.72 kcal·mol^−1^) than lycoramine and galanthamine [50].

The AChE inhibition activity of the species *N. assoanus*, *N. jacetanus*, *N. bujei*, *N. vasconicus*, and *N. pallidulus* have also been described by López and co-authors. However, no activity has been described for the latter two species [41]. Previous publications indicate that the alkaloid composition of the species *N. assoanus* includes assoanine, oxoassoanine, pseudolycorine, 1-*O*-acetylpseudolycorine, and 2-*O*-acetylpseudolycorine [51,52]. The species *N. jacetanus* has been previously documented to contain the compounds assoanine, oxoassoanine, pseudolycorine, and lycorine [53]. According to the literature, the compounds assoanine and oxoassoanine, which belong to a lycorine-type skeleton, are active against AChE, with IC_50_ values of 3.87 ± 0.24 and 47.21 ± 1.13 µM, respectively, while pseudolycorine exhibits poor activity [41]. According to prior publications, the alkaloid lycorine exhibits a very weak in vitro activity against AChE and BuChE, with IC_50_ values higher than 200 µM [41,54]. However, lycorine showed good energy values concerning docking studies with enzymes, with scores of 60.9444 for AChE (1EVE) and 52.7924 for BuChE (homology model) (scores corresponding to galanthamine were 65.4656 and 53.0089) [46].

Cortes et al., 2018 [47] obtained scores of −8.99, −8.87, and −8.94 kcal·mol^−1^ (energy of protein ligand interaction between lycorine and 1DX6 and 4EY7 from AChE, and 4BDS from BuCHE, respectively), while values corresponding to galanthamine were −10.10, −10.20, and −8.23 kcal·mol^−1^, respectively. León et al., 2021 [48] reported binding values for lycorine to AChE (1DX6) and BuChE (4BDS) of −8.82 and −8.94 kcal·mol^−1^, respectively, while values corresponding to galanthamine were −10.10 and −8.23 kcal·mol^−1^. Rojas-Vera et al., 2021 [49], estimated lycorine binding values of −8.89 and −8.38 kcal·mol^−1^ for AChE (4EY7 and 5HF5) and −7.74 kcal·mol^−1^ for BuChE (1P0I), while values corresponding to galanthamine were −9.92, −8.97, and −7.40 kcal·mol^−1^, respectively. Tallini et al., 2018 [54] reported lycorine binding values of −8.82 and −8.94 kcal·mol^−1^ for AchE (1DX6) and BuChE (4BDS), respectively, while scores corresponding to galanthamine were −9.55 and −8.23 kcal·mol^−1^, respectively.

Regarding the molecular modelling of assoanine and oxoassoanine vs. AchE, they did not show great interaction with AchE in terms of electrophilicity, with values of 0.0026 and 0.0036 eV, respectively, (galanthamine electrophilicity values corresponded to 8.5725 eV). In relation to the molecular electrostatic potential (MEP) of assoanine and oxoassoanine, values of negative (−0.08705 and −0.06995 u.a.) and positive (0.03474 and 0.05755 u.a.) regions (which represent the probability to conduct and suffer, respectively, nucleophilic attacks) are not as high as other Amaryllidaceae alkaloids with AChE inhibition properties such as galanthamine (−0.06995 and 0.05755 u.a. negative and positive regions, respectively), hydroxygalanthamine (−0.06822 and 0.06648 u.a.), or sanguinine (−0.07504 and 0.06400 u.a.) [55]. However, it must be considered that no other molecular modeling studies have been found in the literature regarding these two compounds; therefore, more studies should be performed to obtain a global idea of the characterization of the interaction of these compounds with AChE and BuChE.

The alkaloids homolycorine, lycorenine, haemanthamine, 8-*O*-demethylhomolycorine, *O*-methyllycorenine, crinamine, masonine, tazettine, *O*-methyloduline, 11-*O*-acetylhaemanthamine, and bujeine have been documented as constituents of the species *N. bujei* [56]. Additionally, four alkaloids have been reported in the species N. vasconicus, which are vasconine, lycorine, homolycorine, and 8-*O*-acetylhomolycorine [57].

As shown in Table 3, the alkaloid extract of *N. pallidulus* (sample J) was the only extract inactive against both cholinesterases. Previous studies have documented the presence of different structures from the *Sceletium* type in the section Ganymedes (Salisb.) Schultes f. [37,58]. This scaffold is the only group of alkaloids that is not exclusive to the monocotyledon subfamily Amaryllidoideae, being typical structures of the genus *Sceletium* that belongs to the dicotyledonous family Aizoaceae [4,37,59]. According to the literature, the following alkaloids have been described in this plant species: haemanthamine, lycorine, lycorenine, homolycorine, galanthamine, tazettine, mesembrine, mesembrenol, mesembrenone, 2-oxomesembrenone, 7,7a-dehydromesembrenone, 2-oxoepimesembrenol, 6-epimesembrenol, 6-epimesembranol, and 4′-*O*-demethylmesembrenone [58,60].

Considering the inhibitory potential of the samples against both enzymes (Figure 2 and Figure 3), the species *N. confusus*, *N. jonquilla*, and *N. genesii-lopezii* showed remarkable results, with IC_50_ values of 1.04 ± 0.07, 1.88 ± 0.05, and 5.28 ± 0.64 µg·mL^−1^, respectively, against AChE, and 12.83 ± 0.87, 11.72 ± 1.15, and 11.98 ± 0.93 µg·mL^−1^, respectively, against BuChE. López and co-authors [41] evaluated the activity of the alkaloid extract of twenty-six species of *Narcissus* and reported that the best results were obtained for *N. confusus*, which presented a high amount of galanthamine [41]. The literature documents the existence of various structures within the alkaloid composition of the species *N. confusus*, which are ismine, 11,12-dehydroanhydrolycorine, galanthamine, 3-*O*-acetylgalanthamine, *N*-demethylgalanthamine, *N*-formylgalanthamine, narwedine, 8-*O*-methylleucotamine, haemanthamine, haemanthidine, tazettine, pretazettine, 6-*O*-methylpretazettine, epimacronine, homolycorine, 8-*O*-demethylhomolycorine, 9-*O*-demethylhomolycorine, and also narciclasine-, and lycorine-type alkaloids [4,61,62,63]. As indicated in the literature, the chemical diversity of this species presents challenges to its viability as a consistent source of galanthamine on an industrial scale [15].

Various alkaloids have been documented in the alkaloid composition of *N. tazetta*, including galanthamine, sanguinine, narwedine, demethylmaritidine, anhydrolycorine, *O*-methylnorbelladine, pancratinine C, lycorine, 9-*O*-methylpseudolycorine, pseudolycorine, 1-*O*-acetyl-3-*O*-methylnarcissidine, 11, hydroxygalanthine, narcissidine, 9-*O*-demethyl-2alfa-hydroxyhomolycorine, ismine, tazettine, lycorenine, lycorine, masonine, 3-epimacronine, 1,2-dihydroclidanthine, assoanine, hippeastrine, and 4,5-ethylene-8,9-dimethoxy-6-phenanthridone [64,65,66]. Karakoyun and colleagues detailed an analysis of the anticholinesterase potential of alkaloids isolated from the species *N. tazetta* [65]. Their findings revealed 11-hydroxygalanthamine and narcissidine as important compounds with inhibitory effects against AChE, with IC_50_ values of 0.67 and 1.85 µM, respectively, while galanthamine showed IC_50_ values of 0.14 µM [65].

Viladomat and co-authors [67] identified haemanthamine, haemanthidine, tazettine, 3-epimacronine, ismine, and risperidona in the alkaloid composition of *N. asturiensis* [67]. The alkaloid type composition of *N. alcaracensis*, *N. yepesii*, *N. genesii-lopezii*, *N. hedraeanthus*, and *N. jonquilla* was determined by [4]. The alkaloid composition of *N. alcaracensis* and *N. yepesii* is dominated by alkaloid lycorine-type compounds. These species also contain haemanthamine- and homolycorine-type compounds. Pancracine- and galanthamine-type alkaloids have also been reported in *N. yepesii* [4]. *N. genesi-lopezii* composition is dominated by homolycorine-type alkaloids, although the presence of galanthamine-, haemanthamine-, and lycorine-type alkaloids is also reported. The alkaloid composition of *N. hedraeanthus* is dominated by haemanthamine-type alkaloids. However, this species also contains narciclasine-, galanthamine-, tazettine-, and lycorine-type alkaloids [4].

No information about the alkaloid profile of the species *N. minor* and *N. nevadensis* has been found in the literature. Regarding the alkaloid composition and anti-cholinesterase activity of certain species not listed in Table 3, Lisa-Molina and co-workers [6] evaluated the alkaloid profiling of ten samples of *Narcissus*, totaling nine different species (*N. obsoletus*, *N. deficiens*, *N. serotinus*, *N. cavanillesii*, *N. viridiflorus*, *N. elegans*, *N. papyraceus*, *N. bulbocodium*, *N. blancoi*) collected in Spain [6]. The authors detected thirty alkaloids among these samples, with two of them not being identified [6]. According to their results, the lycorine-type skeleton was the most diverse group detected among the samples, with these structures being observed in all of the species, except in *N. viridiflorus* and *N.bulbocodium* [6]. The authors also evaluated the AChE inhibitory activity of ten extracts of *Narcissus*, with seven of them able to act against this enzyme [6]. The best results were observed for the species *N. obsoletus*, which exhibited a substantial concentration of galanthamine in its alkaloid profile and showed IC_50_ values of 0.92 ± 0.06 µg·mL^−1^ [6]. Among all the species evaluated by them, *N. obsoletus* and *N. blancoi* were the only species that showed the presence of alkaloids from the galanthamine-type scaffold [6]. Furthermore, thirteen known and three new Amaryllidaceae alkaloids were isolated from *Narcissus pseudonarcissus* cv. Carlton by Mamun and co-authors [68]. Two of them were named carltonine A and carltonine B. The authors evaluated these new alkaloids in vitro against both cholinesterases, which showed a significant and selective inhibitory activity against BuChE, displaying IC_50_ values of 0.91 ± 0.02 and 0.031 ± 0.001 µM, correspondingly [68]. Twenty-one Amaryllidaceae alkaloids of various structural types and one new alkaloid, named narcimatuline, were obtained from the bulbs of *Narcissus pseudonarcissus* L. cv. Dutch Master by Hulcová and co-workers [69]. According to their results, narcimatuline showed interesting multipotent biological profiling, presenting properties against BuChE, prolyl oligopeptidase (POP), and glycogen synthase kinase-3β (GSK-3β) enzymes, with respective IC_50_ values of 5.9 ± 0.2, 29.2 ± 1.0, and 20.7 ± 2.4 µM [69].

## 4. Conclusions

In Europe, the species *Narcissus* plays a very important role for pharmaceutical companies as a source of galantamine. However, many species of this genus have not yet been studied. In this work, we describe the anticholinesterase potential of fifteen species of *Narcissus* collected in Spain, with seven of them being the first report of their biological activity (*N. minor*, *N. asturiensis*, *N. hedraeanthus*, *N. alcaracensis*, *N. genesii-lopezii*, *N. yepesii* and *N. nevadensis*). Some of these species showed interesting activity against AChE (*N. jacetamus*, *N. assoanus*, *N. minor*, *N. confusus* and *N. jonquilla*) and BuChE (*N. jonquilla*, *N. confusus*, *N. genesii-lopezii*). The species *N. jonquilla* and *N. confusus* show remarkable activity against both enzymes; therefore, they stand out as possible candidates for further studies and for determining their alkaloid composition. This study contributes to underlining the importance of Amaryllidaceae species as a source of important bioactive molecules.

## Figures and Tables

**Figure 1 life-14-00536-f001:**
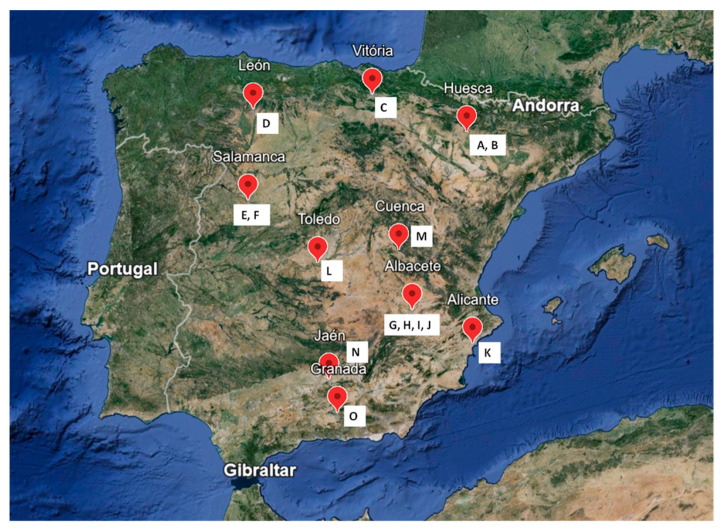
The local collection of the different species of *Narcissus* in Spain according to Table 1 codes. Source: Google Earth. **A** = *N. assoanus*; **B** = *N. jacetanus*; **C** = *N. vasconicus*; **D** = *N. minor*; **E** = *N. confusus*; **F** = *N. asturiensis*; **G** = *N. hedraeanthus*; **H** = *N. alcaracencis*; **I** = N. bujei; **J** = *N. pallidulus*; **K** = *N. tazetta*; **L** = *N. jonquilla*; **M** = *N. genesii-lopezii*; **N** = *N. yepesii*; **O** = *N. nevadensis*. species name when we refer to.

**Figure 2 life-14-00536-f002:**
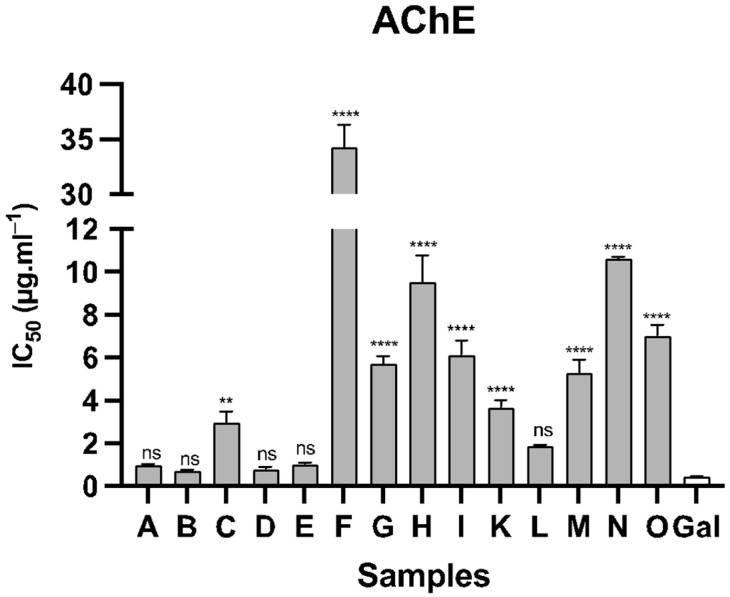
Graph of AChE inhibitory activity of the different samples of alkaloid extracts of *Narcissus* leaves collected in Spain. **A** = *N. assoanus*; **B** = *N. jacetanus*; **C** = *N. vasconicus*; **D** = *N. minor*; **E** = *N. confusus*; **F** = *N. asturiensis*; **G** = *N. hedraeanthus*; **H** = *N. alcaracencis*; **I** = *N. bujei*; **K** = *N. tazetta*; **L** = *N. jonquilla*; **M** = *N. genesii-lopezii*; **N** = *N. yepesii*; **O** = *N. nevadensis*; **Gal** = galanthamine; **** *p* < 0.0001, ** *p* < 0.01, ns—not significant.

**Figure 3 life-14-00536-f003:**
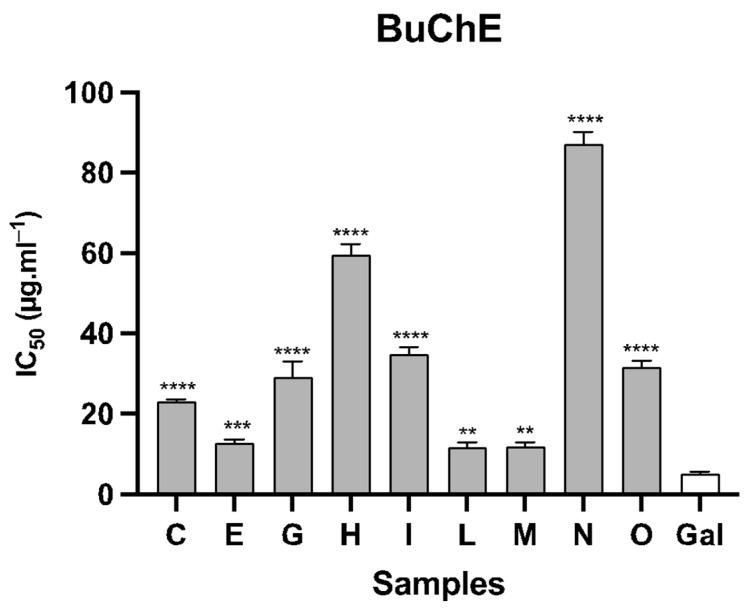
Graph of BuChE inhibitory activity of the different samples of alkaloid extracts of Narcissus leaves collected in Spain. **C** = *N. vasconicus*; **E** = *N. confusus*; **G** = *N. hedraeanthus*; **H** = *N. alcaracencis*; **I** = *N. bujei*; **L** = *N. jonquilla*; **M** = *N. genesii-lopezii*; **N** = *N. yepesii*; **O** = *N. nevadensis*; **Gal** = galanthamine; **** *p* < 0.0001, *** *p* < 0.001, ** *p* < 0.01.

**Figure 4 life-14-00536-f004:**
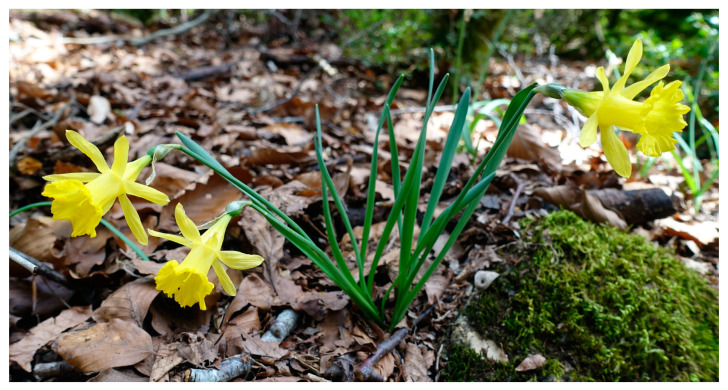
Photo of *Narcissus jacetanus*, the sample with the best results for AChE inhibition.

**Figure 5 life-14-00536-f005:**
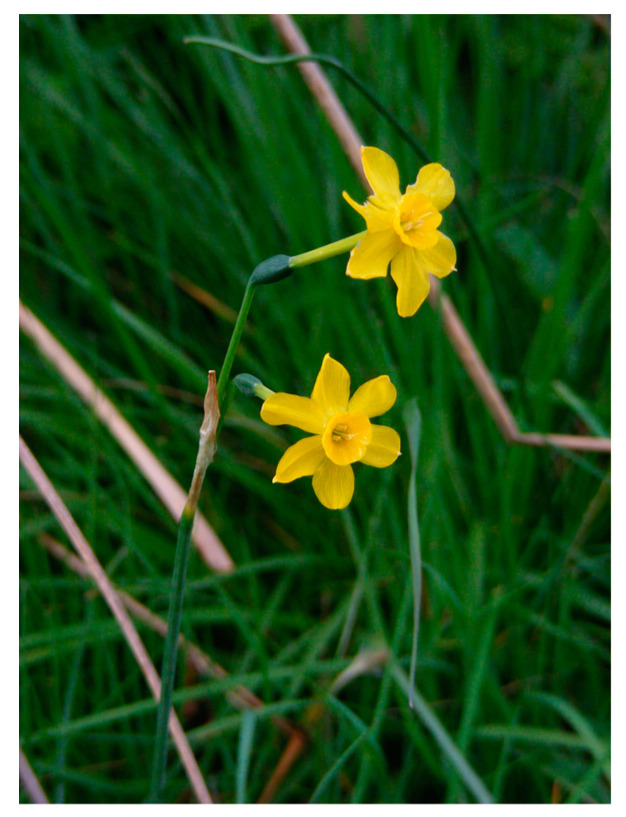
Photo of *Narcissus jonquilla*, the sample with the best results for BuChE inhibition (also with noteworthy activity against AChE).

**Table 1 life-14-00536-t001:** *Narcissus* sample coding, biogeographical distribution, and province of collection. Biogeographical units: Baet: Baetic, CircMed: Circummediterranean, Iber: Iberic, Iber/LAlp: Iberic/Latealpine, Iber/NAfric: Iberic/Northafrican, Med Occ: Mediterranean occidental; political regions of origin: AND: Andalucía, AR: Aragón, CL: Castilla León, CLM: Castilla-La Mancha, VAL: Valencian region, BAS: Basque region. Biogeographical units according to [29].

Code	Species	Section	Biogeographical Distribution	Origin
**A**	*N. assoanus* Dufour ex Schult. and Schult.f.	Jonquillae DC.	Med Occ	Huesca (AR)
**B**	*N. jacetanus* Fern.Casas	Pseudonarcissus DC.	Iber/LAlp	Huesca (AR)
**C**	*N. vasconicus* (Fern.Casas) Fern.Casas	Pseudonarcissus DC.	Iber/LAlp	Vitoria (BAS)
**D**	*N. minor* L.	Pseudonarcissus DC.	Iber/LAlp	León (CL)
**E**	*N. confusus* Pugsley	Pseudonarcissus DC.	Iber/LAlp	Salamanca (CL)
**F**	*N. asturiensis* (Jord.) Pugsley	Pseudonarcissus DC.	Iber/LAlp	Salamanca (CL)
**G**	*N. hedraeanthus* (Webb and Heldr.) Colmeiro	Bulbocodium (Salisb.) DC.	Iber/NAfric	Albacete (CLM)
**H**	*N. alcaracensis* S.Ríos, D.Rivera, Alcaraz and Obón	Nevadensis Zonn.	Baet	Albacete (CLM)
**I**	*N. bujei* (Fern.Casas) Fern.Casas	Pseudonarcissus DC.	Baet	Albacete (CLM)
**J**	*N. pallidulus* Graells	Ganymedes (Salisb.) Schultes f.	Iber	Albacete (CLM)
**K**	*N. tazetta* L.	Tazettae DC.	CircMed	Alicante (VAL)
**L**	*N. jonquilla* L.	Jonquillae DC.	Iber	Toledo (CLM)
**M**	*N. genesii-lopezii* Fern.Casas	Pseudonarcissus DC.	Iber	Cuenca (CLM)
**N**	*N. yepesii* S.Ríos, D.Rivera, Alcaraz and Obón	Nevadensis Zonn.	Baet	Jaén (AND)
**O**	*N. nevadensis* Pugsley	Nevadensis Zonn.	Baet	Granada (AND)

**Table 2 life-14-00536-t002:** Yield of alkaloid extracted, obtained from the *Narcissus* species.

Samples	Dry Weight (g)	Alkaloid Extract (mg)	Yield (%)
**A**	1.00080	8.93	0.89
**B**	1.00114	4.69	0.47
**C**	1.00046	6.05	0.60
**D**	1.00050	4.95	0.49
**E**	1.00032	14.54	1.45
**F**	1.00035	9.79	0.98
**G**	0.88253	6.83	0.77
**H**	1.00069	10.66	1.07
**I**	1.00125	18.83	1.88
**J**	1.00137	10.17	1.02
**K**	1.00103	8.38	0.84
**L**	1.00094	15.32	1.53
**M**	1.00099	6.85	0.68
**N**	1.00037	61.43	6.14
**O**	1.00066	9.27	0.93

**A** = *N. assoanus*; **B** = *N. jacetanus*; **C** = *N. vasconicus*; **D** = *N. minor*; **E** = *N. confusus*; **F** = *N. asturiensis*; **G** = *N. hedraeanthus*; **H** = *N. alcaracencis*; **I** = *N. bujei*; **J** = *N. pallidulus*; **K** = *N. tazetta*; **L** = *N. jonquilla*; **M** = *N. genesii-lopezii*; **N** = *N. yepesii*; **O** = *N. nevadensis*.

**Table 3 life-14-00536-t003:** IC_50_ values of AChE and BuChE inhibitory activity of the Narcissus species collected in Spain. Values expressed in µg·mL^−1^. **A** = *N. assoanus*; **B** = *N. jacetanus*; **C** = *N. vasconicus*; **D** = *N. minor*; **E** = *N. confusus*; **F** = *N. asturiensis*; **G** = *N. hedraeanthus*; **H** = *N. alcaracencis*; **I** = *N. bujei*; **K** = *N. tazetta*; **L** = *N. jonquilla*; **M** = *N. genesii-lopezii*; **N** = *N. yepesii;*
**O** = *N. nevadensis*; **Gal** = galanthamine.

Samples	Species	AChE	BuChE
**A**	*N. assoanus*	0.99 ± 0.06	>100
**B**	*N. jacetanus*	0.75 ± 0.03	>100
**C**	*N. vasconicus*	2.98 ± 0.51	23.14 ± 0.58
**D**	*N. minor*	0.81 ± 0.10	>100
**E**	*N. confusus*	1.04 ± 0.07	12.83 ± 0.87
**F**	*N. asturiensis*	34.28 ± 2.06	>100
**G**	*N. hedraeanthus*	5.73 ± 0.36	29.23 ± 3.84
**H**	*N. alcaracencis*	9.54 ± 1.26	59.60 ± 2.61
**I**	*N. bujei*	6.14 ± 0.68	34.86 ± 1.75
**J**	*N. pallidulus*	>100	>100
**K**	*N. tazetta*	3.68 ± 0.36	>100
**L**	*N. jonquilla*	1.88 ± 0.05	11.72 ± 1.15
**M**	*N. genesii-lopezii*	5.28 ± 0.64	11.98 ± 0.93
**N**	*N. yepesii*	10.63 ± 0.08	87.20 ± 3.01
**O**	*N. nevadensis*	7.03 ± 1.49	31.68 ± 0.49
**Gal**		0.46 ± 0.03	5.13 ± 0.48

Gal: galanthamine.

## Data Availability

Data are contained within the article.

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
