# Peer review of "The Anti-Cholinesterase Potential of Fifteen Different Species of *Narcissus* L. (Amaryllidaceae) Collected in Spain"

_life, 2024, doi:10.3390/life14040536_

Round 1
Reviewer 1 Report
Comments and Suggestions for Authors
The manuscript entitled " Anti-cholinesterase potential of fifteen different species of Narcissus L. (Amaryllidaceae) collected in Spain" is well written. This study has great importance regarding the pharmaceutical industry.
Author Response
We are deeply grateful for reviewer’s comments and highly appreciate the positive assessment.
Reviewer 2 Report
Comments and Suggestions for Authors
The manuscript studied Anti-choline esterase of plants from the genus Narcissus L.
The species N. jacetanus exhibited the best inhibition values against AChE, with IC50 values of 0.75 µg·ml-1, while Narcissus jonquilla was the most active against BuChE, with IC50 values of 11.72 µg·ml-1
The authors should reveal the role of previously identified alkaloids as anticholine esterase through molecular modelling or re isolation of these compound and evaluate each compound alone against choline esterase.
Therefore, the manuscript should not accepted at its present form.
Author Response
We have added two new paragraphs (line 296 to 303, and line 316 to 325) with the molecular modelling studies with AChE and BuChE of such alkaloids.
We have added 6 references with data required by the reviewer:
Bozkurt, B.; Coban, G.; Kaya, G.I.; Onur, M.A.; Unver-Somer, N. Alkaloid profiling, anticholinesterase activity and molecular docking of Galanthus elwesii. S Afr J Bot 2017, 113, 119-127. https://doi.org/10.1016/j.sajb.2017.08.004
Cortes, N.; Sierra, K., Alzate, F.; Osorio, E.H.; Osorio, E. Alkaloids of Amaryllidaceae as Inhibittors of Cholinesterases (AChE and BuChEs): An Integrated Bioguide Study. Phytochem Anal 2017, 29(2), 217-227. https://doi.org/10.1002/pca.2736
León, K.A.; Inca, A.; Tallini, L.R.; Osorio, E.H.; Robles, J.; Bastida, J.; Oleas, N.H. Alkaloids of Phaedranassa dubia (Kunth) J.F. Macbr. and Phaedranassa brevifolia Meerow (Amaryllidaceae) from Ecuador and its cholinesterase-inhibitory activitty. S Afr J Bot 2021, 136, 91-99. https://doi.org/10.1016/j.sajb.2020.09.007
Rojas-Vera, J. de C.; Buitrago-Díaz, A.A.; Possamai, L.M.; Timmers, L.F.S.M.; Tallini, L.R.; Bastida, J. Alkaloid profile and cholinesterase inhibition activity of five species of Amaryllidaceae family collected from Mérida state-Venezuela. S Afr J Bot 2021, 136, 126-136. https://doi.org/10.1016/j.sajb.2020.03.001
Tallini, L.R.; Osorio, E.H.; Berkov, S.; Torras-Claveria, L.; Rodríguez-Escobar, M.L.; Viladomat, F.; Meerow, A.; Bastida, J. Chemical Survey of Three Species of the Genus Rauhia Traub (Amaryllidaceae). Plants 2022, 11, 3549. https://doi.org/10.3390/plants11243549
Brito, M. de F. de; Ferreira, J.V.; de Souza, L.R.; Gemaque, L.R.P.; Sousa, K.P.A.; dos Santos, C.F.; Braga, F.S.; Pernomian, L.; da Silva, C.H.T.P.; Santos, C.B.R.; Taft, C.A.; Hage-Melim, L.I.S. Computational Molecular Modeling of Compounds from Amaryllidaceae Family as Potential Acetylcholinesterase Inhibitors. Current Bioactive Compounds 2017, 13(2), 121-129. https://doi.org/10.2174/1573407212666160607093830
Reviewer 3 Report
Comments and Suggestions for Authors
The purpose of this paper is to evaluate the potential of these plants in Alzheimer’s disease.and it is generally recommended for publication once the following issues listed below are carefully addressed.
1. Characterization is critical for the discovery of bioactive ingredients and extracts, yet some important data regarding the characterizations of the extracts and alkaloids are missing, along with information on how the alkaloids were identified. I highly suggest supplementing the characterization of the alkaloids. Additionally, information on yield and purity is needed.
2. In the preparation of alkaloid extracts, some data are missing again and should be added. This includes the yield of the extracts obtained from each of the plants, the yield of the alkaloids, the volume of solvents used for extraction, and the raw materials used for the extraction process.

Author Response
1.
Narcissus plants are a valuable resource for pharmaceutical companies because they contain galanthamine, a key ingredient in certain medications for Alzheimer’s disease. In Europe, Narcissus pseudonarcissus is the principal source of galanthamine by pharmaceutical companies. Surprisingly, there haven't been many studies on the anti-cholinesterase potential of different Narcissus species available in literature., we want to contribute to this area of research. Therefore, we have plan to investigate the anti-cholinesterase potential of several Narcissus species. The characterisation of bioactive components of each plant species is a very interesting objective for the future as a continuation of the present research. In fact, given our extensive experience with Amaryllidaceae alkaloids (the plant family that includes Narcissus) this is one of our forthcoming goals. It has to be taken in account that the characterisation of bioactive compounds entails a meticulous and highly specific work, that requires several specialised techniques and software tools, and that it will yield a substantial volume of information and data, which may extend beyond the scope of the present paper. The current study serves as a preliminar screening of selected wild Narcissus species, focusing on their activity inhibiting AChE and BuChE, which are implicated in Alzheimer's disease treatment. The determination of all alkaloids identity will, then, constitute the following step, that will offer a broader spectrum of insights and information. While some of these alkaloids may not exhibit activity in inhibiting AChE and BuChE, others may possess diverse biological activities such as antiparasitic, antibacterial, or antiviral, justifying a single careful examination.
2.-
The missing data, including the yield of the extracts obtained from each plant, the yield of the alkaloids, the volume of solvents used for extraction, and the raw materials utilized in the extraction process have been added. These data are included in the section 2.2 and in the lines 240 to 244. Also, Table 2 has been added in line 246 regarding this information.
Round 2
Reviewer 2 Report
Comments and Suggestions for Authors
The authors should write the score of docking to clarify the potency of the compounds.
Author Response
Docking scores have been added (lines 298 to 307, and 320 to 349).
Reviewer 3 Report
Comments and Suggestions for Authors
Accept in present form. Thanks
Author Response
We are grateful for reviewer’s comment and appreciate the positive assessment.